# SynHING: Synthetic Heterogeneous Information Network Generation for Graph Learning and Explanation

## Abstract

Graph Neural Networks (GNNs) excel in modeling graph structures across diverse domains, such as community analysis and recommendation systems. As the need for GNN interpretability grows, there is an increasing demand for robust baselines and comprehensive graph datasets, especially within the realm of Heterogeneous Information Networks (HIN). To address this, we introduce SynHING, a framework for Synthetic Heterogeneous Information Network Generation designed to advance graph learning and explanation. After identifying key motifs in a target HIN, SynHING systematically employs a bottom-up generation process with intra-cluster and inter-cluster merge modules. This process, along with post-pruning techniques, ensures that the synthetic HIN accurately mirrors the structural and statistical properties of the original graph. The effectiveness of SynHING is validated using four datasets - IMDB, Recipe, ACM, and DBLP - spanning three distinct application categories, demonstrating both its generality and practicality. Furthermore, SynHING provides ground-truth motifs for evaluating GNN explainer models, establishing a new benchmark for explainable, synthetic HIN generation. This contributes significantly to advancing interpretable machine learning in complex network environments.

## 1 Introduction

In recent years, Graph Neural Networks (GNNs) have shown impressive performance in various graph analysis tasks such as community analysis (Shchur & Günnemann, 2019), chemical bond analysis (Stokes et al., 2020), and recommendation systems (Cui et al., 2020). These tasks include node and edge classification, link prediction, and clustering (Chami et al., 2022). Heterogeneous Information Networks (HINs) consist of multiple types of nodes and edges that contain various information, providing a natural representation of real-world data. The development of Heterogeneous Graph Neural Networks (HGNN) has been sparked by HINs. These networks can be classified into different types of models, including meta-path-based models like HAN (Wang et al., 2019) and MAGNN (Fu et al., 2020), transformer-based GNN models (Yun et al., 2020; Hu et al., 2020b), as well as SimpleHGN (Lv et al., 2021) which uses projection layers to map information to a shared feature space and aggregates information using an edge-type attention mechanism. Additionally, TreeXGNN (Hong et al., 2023) combines a tree-based feature extractor with the HGNN model to enhance performance. However, due to the lack of public HIN datasets, most existing HGNNs are trained and evaluated on only a few known public datasets such as IMDB [1], ACM (Wang et al., 2019), and DBLP [2], leading to potential bias. The scarcity of public HIN datasets compared to other machine learning domains poses significant challenges for HGNNs, potentially causing overfitting and hindering effective generalization (Palowitch et al., 2022).

Due to the scarcity of HIN datasets, it is even more difficult to study trustworthy and interpretable models. As trust, transparency, and privacy become essential for machine learning models, it is important to reveal the decision-making process and knowledge hidden behind models. Most existing GNN models lack transparency, making them difficult to be trusted and limiting their applicability in

---

[1] https://www.kaggle.com/datasets/karrrimba/movie-metadatacsv
[2] http://web.cs.ucla.edu/~yzsun/data/

decision-critical scenarios. GNN explainer models were proposed to disclose the black box. Nevertheless, there is no objective way to evaluate the performances of GNN explainers as there exists no suitable dataset in public domains that provides the data, objective, and corresponding ground-truth explanations at the same time.

Synthetic homogeneous graph datasets have been introduced (Ying et al., 2019), attempting to alleviate the lack of explainable datasets. These datasets are created by randomly attaching specially designed structured network motifs to artificial graphs (Albert & Barabási, 2002). These motifs are then utilized as ground-truth explanations. There have been very few attempts to generalize synthetic graph generation to HINs. This generalization is non-trivial for two reasons: Firstly, the basic artificially structured motifs, such as houses and grids (Ying et al., 2019), common in the synthesis of homogeneous graphs, are inadequate for HINs. These simplistic motifs often fail to capture the complexity and diversity of real-world heterogeneous graphs, resulting in a poor representation of the intricate structures found in actual HIN environments. Secondly, methods designed for generating homogeneous synthetic graphs cannot be directly applied to heterogeneous information networks (HINs) due to their unique structural constraints. Traditional approaches, which often involve randomly connecting nodes to form edges, may result in illegitimate connections within HINs. Such connections can violate the semantic rules inherent to HINs, where edges must align with the types of nodes they connect to represent accurate and meaningful relationships. Moreover, these methods randomly add edges, merely increasing node degrees without considering the global structure of the graph. This randomization disrupts the alignment of node degree and structural patterns with real-world HINs. Additionally, as highlighted by Li et al. (2023a), the scarcity of heterogeneous datasets with ground-truth explanations further complicates the development of explanation models for HINs. Researchers often resort to indirect evaluation methods, which may not effectively capture the true performance of these models.

In light of the above concerns, we introduce Synthetic Heterogeneous Information Network Generation (SynHING), a novel framework that constructs synthetic HINs for graph learning and explanation by referencing existing real graphs. SynHING methodically generates major motifs essential for explanations and employs our newly developed Intra-/Inter-cluster Merge method. This method facilitates the merging of multiple subgraph groups to systematically create synthetic HINs of any specified size. Additionally, we introduce the concept of exclusion that allows us to flexibly adjust the complexity of node prediction tasks within these synthetic graphs while maintaining structural similarity to the reference graph. This feature enhances the utility of SynHING in practical applications. The synthetic graphs generated by SynHING provide interpretable insights, aiding significantly in the explanation tasks associated with HINs. SynHING has been validated using four datasets: IMDB, Recipe, ACM, and DBLP, which cover three distinct application categories, demonstrating its generality and practicality. Furthermore, theoretical complexity analysis has been conducted to showcase its scalability. [3]

## 2 RELATED WORK

### 2.1 SYNTHETIC GRAPH GENERATION

Artificially synthesized data has a long history of development (Kingma & Welling, 2013; Bowyer et al., 2011; Dong et al., 2018; Frid-Adar et al., 2018; Karras et al., 2019; Xu et al., 2019; Figueira & Vaz, 2022). With the growth of Graph Neural Networks (GNN), there has been a renewed interest in synthetic graph generation algorithms. Early on, Snijders & Nowicki (1997) introduced a method to generate graph edges based on node clusters. More contemporary approaches, such as those described by Dwivedi et al. (2020), extract subgraphs from real-world graphs to test GNNs' ability to identify specific substructures within Stochastic Block Model (SBM) graphs. The concept of SBM, particularly popular for generating clusters that maintain strong intra-node correlations, has been adapted into various forms, including unsupervised (Tsitsulin et al., 2020) and semi-supervised models (Rozemberczki et al., 2021). Further extending SBM's utility, GraphWorld (Palowitch et al., 2022) leverages the Degree-Corrected Stochastic Block Model (DC-SBM) (Abbe, 2017) to create diverse graph datasets using multiple parameters. Yet, these works mainly focus on homogeneous

---

[3]Open-source code will be released upon acceptance.

graphs, and most synthetic graph generation methods do not provide the ability for interpretation, lack ground-truth explanations, and cannot be directly extended to HINs.

## 2.2 EXPLAINER FOR GRAPH NEURAL NETWORKS

Developing trustworthy machine learning models is now a widely acknowledged goal within the community. Explainability methods for Graph Neural Networks (GNNs) have seen considerable development, especially for node and graph classification tasks (Ying et al., 2019; Luo et al., 2020; Yuan et al., 2021; Lin et al., 2022). These methods generally fall into two categories: inherent interpretable models and post-hoc explainability approaches (Yuan et al., 2020). Inherent interpretable models, such as ProtGNN (Dai & Wang, 2021), integrate explanations directly within the model, using mechanisms like top-K similarity to identify influential subgraphs for their predictions. In contrast, post-hoc explainability methods focus on identifying crucial subgraphs (Luo et al., 2020; Yuan et al., 2021) and key features (Ying et al., 2019), approximating the behavior of black-box models by elucidating the connections between inputs and outputs. While most data validating these explainer models are derived from homogeneous graphs with simplistic motifs (Ying et al., 2019), such as houses and grids, these often lack the diversity and complexity of real-world graphs. Some researchers are studying heterogeneous GNN explanations (Li et al., 2023b; Lv et al., 2023), but the lack of HINs with explanation ground truths poses challenges. Consequently, generating appropriate heterogeneous graph datasets for validation is emerging as a crucial research area.

## 2.3 DATASETS WITH GROUND-TRUTH EXPLANATIONS

The conventional evaluation of graph explanatory methods often relies on molecular datasets or synthetic community node classification datasets. These datasets are favored because they offer ground-truth explanations. For instance, MUTAG is a molecular dataset containing graphs labeled according to their mutagenic effect (Debnath et al., 1991). Synthetic datasets, as introduced by Ying et al. (2019), are node classification datasets created by randomly attaching structured network motifs to base graphs. These motifs are designed with specific structures such as houses, cycles, and grids. The base graphs are generated either through BA methods or as a balanced binary tree. However, these datasets are homogeneous and do not transition well to heterogeneous graph contexts. In contrast, our approach involves extracting relevant motifs directly from actual HINs to serve as the basis for ground-truth explanations and employing a systematic bottom-up method to guarantee inherent semantic rules of HINs when generating HIN datasets.

## 3 PROPOSED METHOD: SYNHING

### 3.1 PRELIMINARIES

HINs, also called heterogeneous graphs, consist of multiple node and edge types, which can be defined as a graph $G = (\mathcal{V}, \mathcal{E}, \Phi, \Psi)$, where $\mathcal{V}$ and $\mathcal{E}$ is the set of nodes and edges, respectively. Each node $v \in \mathcal{V}$ has a type $\Phi(v) \in \mathcal{T}_v$, and each edge $e \in \mathcal{E}$ has a type $\Psi(e) \in \mathcal{T}_e$. The node feature matrix is denoted by $F_\phi \in \mathbb{R}^{|\mathcal{V}^\phi| \times d_\phi}$, where $\mathcal{V}^\phi$ is the set of node with node type $\phi$, i.e. $\mathcal{V}^\phi = \{ v \in \mathcal{V} \mid \Phi(v) = \phi \}$, and $d_\phi$ is the feature dimension of the node type $\phi$. Target nodes of the graph $G$, denoted by $\mathcal{V}^{\phi_0}$, are associated with labels collected as $Y \in \mathcal{Y}^{|\mathcal{V}^{\phi_0}|}$, where $\phi_0 \in \mathcal{T}_v$ denotes the target node type.

### 3.2 OVERVIEW OF SYNHING

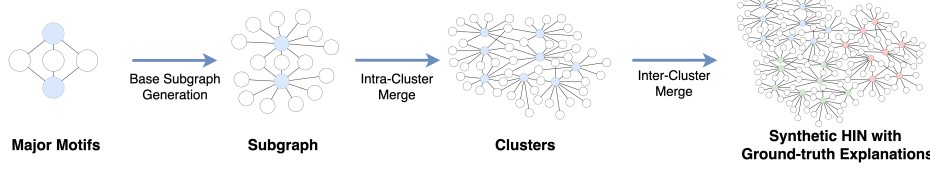

Figure 1: Synthetic HIN Generation Flow

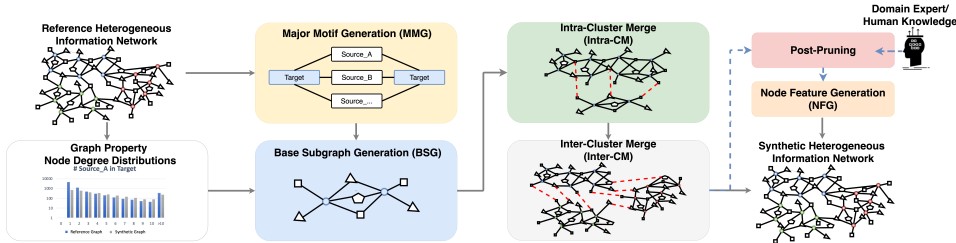

Figure 2: SynHING

In this study, we proposed SynHING, a novel synthetic HIN generation framework. We aim to generate an arbitrary size synthetic heterogeneous graph $\tilde{G}$ with the explanation ground truths, which closely mimics the property of the given real-world graph $\hat{G}$ through a bottom-up generation process, which is demonstrated in Fig. 1. Firstly, we generate the major motifs and derive base subgraphs from them with proper node degree distributions. Secondly, we introduce two merge modules to handle partial graphs; Intra-Cluster Merge aims to merge subgraphs within a cluster, and Inter-Cluster Merge is used to merge clusters with different labels. Finally, the synthetic HIN will be generated after feature generation and post-pruning. SynHING framework consists of six modules, as shown in Fig. 2: (1) Major Motif Generation, (2) Base Subgraph Generation, (3) Intra-Cluster Merge, (4) Inter-Cluster Merge, (5) Node Feature Generation, and (6) Post-Pruning. The details of the proposed methods will be introduced in the following sections.

### 3.3 MAJOR MOTIF GENERATION (MMG)

To generate major motifs for ground-truth explanations, we identify meta-paths within the referenced graph that originate and terminate with target nodes, with all intermediate nodes. The major motif is derived by designating two target nodes as anchors and connecting them through all possible meta-paths within a specified number of hops (Wang et al., 2019; Fu et al., 2020). The maximum number of hops can be set manually or based on the number of layers in the GNN models, reflecting that GNN computation graphs are represented by $n$-hop subgraphs, where $n$ matches the model layers. As depicted in Fig. 2, the MMG module utilizes three one-hop paths to form a major motif as an example.

Further investigation revealed that these derived motifs are common patterns found using the graphlet searching method (Milo et al., 2002) in real-world graphs. We conducted experiments on the IMDB dataset, identifying one of the most common graphlets, $G20$ (Milo et al., 2002), which features multiple minor nodes acting as bridges between two target nodes. A similar pattern is noted in Megnn (Chang et al., 2022). We define these robust graph patterns as the major motif, providing essential ground truths for explanations. In addition, the motif can also be defined by the user and customized for diverse explanation tasks.

### 3.4 BASE SUBGRAPH GENERATION (BSG)

Based on the major motifs previously generated, we further develop base subgraphs. In this base subgraph generation process, we introduce randomness into the major motifs and augment them with several non-target nodes, designated as minor nodes, attached to target nodes within each motif. This addition aims to create diverse subgraphs with noise, which are not part of the ground truths for explanations but mimic the real-world reference graph.

These minor nodes fulfill two primary functions. Firstly, they help match the degree distributions of the target nodes in the subgraphs to those observed in the referenced real-world distribution, denoted as $P^\phi(k)$, where $k$ is the number of connections to nodes of type $\phi$. Secondly, the minor nodes serve as crucial junction points for subsequent merging processes.

Once minor nodes are added, each pair of target nodes within a motif is assigned identical labels, defining this entity as a *base subgraph*. This is denoted as a tuple $(S_i, y_i)$, where $S_i = (\mathcal{V}_i, \mathcal{E}_i)$ represents the structure of the $i$-th subgraph, and $y_i \in \mathcal{Y}$ is the label of the associated target nodes. The generated subgraphs are then collected into sets $\mathcal{K}_y$ for each label $y \in \mathcal{Y}$.

## 3.5 MERGE TO GENERATE HINs

Conventional methods for constructing graphs often involve adding edges between nodes or subgraphs to create a connected homogeneous graph. However, this approach carries the risk of inadvertently forming illegal connections despite careful node selection to prevent them. It is also challenging to maintain statistical properties, such as node degree distributions relative to different node types, when connecting nodes.

To overcome these challenges, we propose a novel *Merge* operation that combines two nodes into one, which will connect back to the initial neighbors of the two nodes. This method elegantly adheres to existing constraints and preserves the degree distributions of the target nodes within the generated subgraphs, ensuring an accurate representation of the target real-world graph. The general merge function operates as follows: Given a graph $G = (\mathcal{V}, \mathcal{E})$ and two nodes $v_1, v_2 \in \mathcal{V}$. If we merge $v_1$ and $v_2$ into $v_1$ in $G$, then the process is defined by

$$(\mathcal{V}', \mathcal{E}') = \text{Merge}(v_1, v_2; G) \tag{1}$$

$$= (\mathcal{V} \setminus \{ v_2 \}, \mathcal{E} \cup \{ (v_1, u) \mid u \in N(v_2), u \neq v_1 \} \setminus \{ (v_2, u) \mid u \in N(v_2) \}), \tag{2}$$

where $N(v)$ represents the neighbors of the node $v$. The *Merge* operation connects the neighbors of node $v_2$ to node $v_1$ while simultaneously removing the merged node $v_2$ and its original edges to its neighbors. We use $\text{Merge}(\mathfrak{P}; G)$ to denote merging multiple pairs in $\mathfrak{P} \subseteq \mathcal{V} \times \mathcal{V}$ in $G$ (note that the order in $\mathfrak{P}$ does not matter). Similarly, $\text{Merge}(\mathfrak{P}; G_1 \oplus G_2 \oplus \dots)$ signifies the merging of pairs across multiple graphs $G_1, G_2, \dots$, where $\oplus$ denotes the graph disjoint union operator. Next, we generate a complete synthetic HIN from bottom to top with Intra-/Inter-Cluster Merges.

### 3.5.1 INTRA-CLUSTER MERGE (INTRA-CM)

Intra-CM is designed to merge the subgraphs with identical labels one by one to form a cluster denoted by $C_y = (\mathcal{V}_y, \mathcal{E}_y)$, which can mirror the "Superstar" phenomenon often observed in community networks (Albert & Barabási, 2002; Abbe, 2017). In social networks, early adopters often become central nodes or "Superstars" due to the sequential nature of network growth. As new members join, they tend to connect with already well-established individuals. Early joiners accumulate more connections over time, benefiting from the principle of preferential attachment, where new links are more likely to form with highly connected members. This process leads to a few early adopters amassing a disproportionate number of connections, thereby becoming key influencers or opinion leaders within the network.

In brief, for each label $y$, we denote $\mathcal{K}_y$ as the set of base subgraphs of the same label generated earlier in the BSG step. These base subgraphs are sequentially merged into the cluster $C_y$. The entire Intra-CM process is conducted $|\mathcal{Y}|$ times to produce all clusters $\{ C_y \mid y \in \mathcal{Y} \}$ for all labels. Note that we form each cluster independently. Since different node types need to be handled carefully in HINs, Intra-CM is conducted separately for each node type, denoted by $\phi$. Specifically, the initial subgraph $S_0$ is chosen from $K_y$ to be the original cluster $C_y^0$. At each iterations $i$, we select a subgraph $S_i = (\mathcal{V}_i, \mathcal{E}_i)$ from the remaining $\mathcal{K}_y \setminus \{ S_0, ..., S_{i-1} \}$ and merge $C_y^{i-1}$ and $S_i$ to generate $C_y^i$. For each minor node type $\phi \neq \phi_0$, we initially determine the number of *Merge* operations to be performed, denoted as $n_{\text{intra}}^\phi$. The number $n_{\text{intra}}^\phi$ is sampled from a binomial distribution, i.e.

$$n_{\text{intra}}^\phi \sim \text{B}(n = |\mathcal{V}_i^\phi|, p = p^\phi), \tag{3}$$

where $p^\phi$ is the Intra-CM probability for the minor node type $\phi$. A higher Intra-CM probability leads to more nodes being merged during this step, resulting in a tighter connection graph within the cluster, which means increased exclusion between clusters. To merge nodes within $S_i$ and $C_y$, we need to identify the sampling space of the node pairs:

$$\mathfrak{M}_{\text{intra}}^\phi = \{ \{ v_y, v_i \} \mid v_y \in \mathcal{V}_y^\phi, v_i \in \mathcal{V}_i^\phi \}. \tag{4}$$

Subsequently, $n_{\text{intra}}^\phi$ pairs are then sampled uniformly from $\mathfrak{M}_{\text{intra}}^\phi$ into $\mathfrak{P}^\phi \subseteq \mathfrak{M}_{\text{intra}}^\phi$ without replacement. The *Merge* operation is performed on the sampled pairs of all minor node types to merge the $S_i$ into the cluster:

$$C_y^i = \text{Merge} \left( \bigcup_{\phi \in \mathcal{T}_v, \phi \neq \phi_0} \mathfrak{P}^\phi; \quad C_y^{i-1} \oplus S_i \right), \tag{5}$$

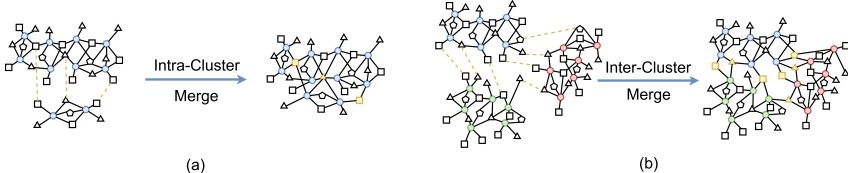

Figure 3: Intra-Cluster and Inter-Cluster Merges

where $\bigcup$ denotes the union over all minor node types. After the above Intra-CM, we can generate multiple clusters with all labels, shown in Fig. 3a.

### 3.5.2 INTER-CLUSTER MERGE (INTER-CM)

Inter-CM aims to merge clusters with different labels. Unlike intra-clusters, the "Superstar" phenomenon should not occur in inter-clusters because clusters would not appear sequentially to form the whole graph. Hence, we should concurrently merge all clusters rather than sequentially merge subgraphs, as in Intra-CM. In short, the proposed Inter-CM merges node pairs from different clusters to form the complete graph structure $\tilde{G}$, using clusters $\{ C_y \mid y \in \mathcal{Y} \}$ produced by Intra-CM.

Analogous to Intra-CM, all merges need to be conducted separately for each node type $\phi$. For each node type $phi$, the initial step in Inter-CM involves identifying potential pairs $\mathfrak{M}_{\text{inter}}^{\phi}$ to be merged, formally defined as follows:

$$\mathfrak{M}_{\text{inter}}^{\phi} = \left\{ \{ v_1, v_2 \} \mid v_1 \in \mathcal{V}_{y_1}^{\phi}, \, v_2 \in \mathcal{V}_{y_2}^{\phi}, \, \{ y_1, y_2 \} \subseteq \mathcal{Y}, y_1 \neq y_2 \right\}, \tag{6}$$

where $\mathcal{V}_{y_1}^{\phi}, \mathcal{V}_{y_2}^{\phi}$ are nodes in type $\phi$ of two different clusters $C_{y_1}, C_{y_2}$, respectively. The number of pairs $n_{\text{inter}}^{\phi}$ is sampled from a binomial distribution:

$$n_{\text{inter}}^{\phi} \sim B \left( n = \sum_{y \in \mathcal{Y}} |\mathcal{V}_y^{\phi}|, k = q^{\phi} \right), \tag{7}$$

where $q^{\phi}$ is the Inter-Cluster merge probability. A higher value of $q^{\phi}$ results in more nodes from different clusters being merged, leading to a graph with lower exclusion of clusters. The $n_{\text{inter}}^{\phi}$ pairs are sampled uniformly from $\mathfrak{M}_{\text{inter}}^{\phi}$ to form the set of node pairs that we intend to merge $\mathfrak{P}^{\phi}$. The clusters are merged based on $\mathfrak{P}^{\phi}$ and form a complete graph $\tilde{G}$:

$$\tilde{G} = \text{Merge} \left( \bigcup_{\phi \in \mathcal{T}_v'} \mathfrak{P}^{\phi}; \quad \bigoplus_{y \in \mathcal{Y}} C_y \right), \tag{8}$$

where $\bigoplus$ denotes the graph disjoint union over all labels $y \in \mathcal{Y}$. If the generated graph $\tilde{G}$ is intended to be multi-label, $\mathcal{T}_v' = \mathcal{T}_v$, i.e., all node types, including target node type $\phi_0$, are allowed to be merged. Conversely, in the case of single-label, merging target nodes is not permitted, i.e., $\mathcal{T}_v' = \mathcal{T}_v \setminus \{ \phi_0 \}$. After the above Inter-CM, we can generate a complete heterogeneous graph structure with multiple labels, as shown in Fig. 3b.

### 3.6 NODE FEATURE GENERATION (NFG)

For NFG, we sample node features from within-cluster multivariate normal distributions, following previous studies(Palowitch et al., 2022; Tsitsulin et al., 2022). The node features in the same cluster will be sampled from a shared prior multivariate normal distribution $\mathcal{N}(\mu_y, \alpha)$, where $\mu_y$ represents the feature center sampled from another normal distribution $\mu_y \sim \mathcal{N}(0, \beta)$. Here, $\alpha/\beta$ serves as a hyperparameter, representing the ratio of feature center distance to cluster covariance. It can be interpreted as a signal-to-noise ratio (SNR). For the multi-label nodes, we generate node features based on a joint probability distribution that combines multiple independent probability distribution functions. Afterward, we draw samples from this combined distribution to determine the features of the nodes. In this work, we generated node features for target nodes because minor node features are

often unavailable in the real-world heterogeneous graphs and are commonly preprocessed by either constants, node IDs, or propagated features (Lv et al., 2021). Therefore, we follow this common setting, using node ID or node type information as node features for minor nodes to approximate these real-world datasets.

### 3.7 POST-PRUNING (P-P)

P-P is an optional yet critical process applied to synthetic graphs to ensure they adhere to constraints observed in raw data. For instance, in the IMDB dataset, each movie is linked to no more than three actors, reflecting inherent limits within the original dataset. During P-P, we first establish the upper limits of node degrees and scan the HIN node-by-node to remove excess edges until the node degrees conform to the upper limit constraints. Importantly, we prioritize the retention of edges that form part of the major motif, thereby preserving the integrity of the explanation ground truths.

### 3.8 COMPLEXITY OF SYNHING

To explore SynHING's scalability, we conducted a complexity analysis. The process of MMG and BSG are both independent. Therefore, the time complexity of MMG and BSG is $O(N)$. The complexity of Intra-CM is $O(N|\mathcal{V}_i| + N|\mathcal{E}_i|)$ or $O(N)$, as $|\mathcal{V}_i|$ and $|\mathcal{E}_i|$ are the number nodes and edges in the base subgraph, which are constant w.r.t. $N$. The processes involved in Inter-CM are similar to Intra-CM; the complexity is also $O(N)$. Therefore, the overall time complexity of SynHING is determined by the number of motifs $N$ with a time complexity of $O(N)$, which showcases the scalability of SynHING. More details can be found in the Appendix A.1.

## 4 EXPERIMENTAL SETTINGS

### 4.1 DATASETS AND HGNNS

To assess the SynHING framework, we generate synthetic graphs using four well-established HIN node classification datasets: IMDB 1, Recipe (Majumder et al., 2019), ACM (Wang et al., 2019), and DBLP 2. Fig. 4a presents the graph schema for these four heterogeneous datasets that illustrate the permissible edge types within each graph.

To identify major motifs, we designate two target nodes as anchors and connect them using all feasible meta-paths, limited by a specified number of hops. Specifically, we utilize two hops for the IMDB, Recipe, and ACM datasets and four hops for the DBLP dataset to align with their respective graph schemas, as depicted in Fig. 8. We further study the minor node degree of the approximate reference graph and real graph. More details can be found in the Appendix A.7.

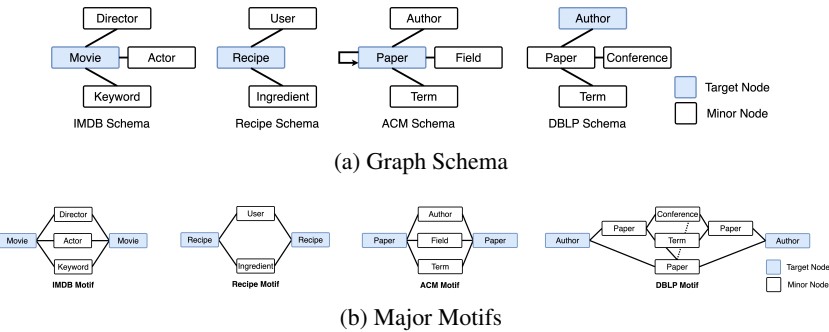

(a) Graph Schema

(b) Major Motifs

Figure 4: Graph Schema and Major Motifs of the Four Heterogeneous Graph Datasets

We utilize the transductive learning approach for node classification tasks and randomly select 24% of the target nodes for training, 6% for validation, and 70% for testing (Wang et al., 2019; Lv et al., 2021; Hong et al., 2023). We used three well-known HGNNs, HGT (Hu et al., 2020b), SimpleHGN (Lv et al., 2021), and TreeXGNN (Hong et al., 2023), as our encoders to validate the synthetic HINs.

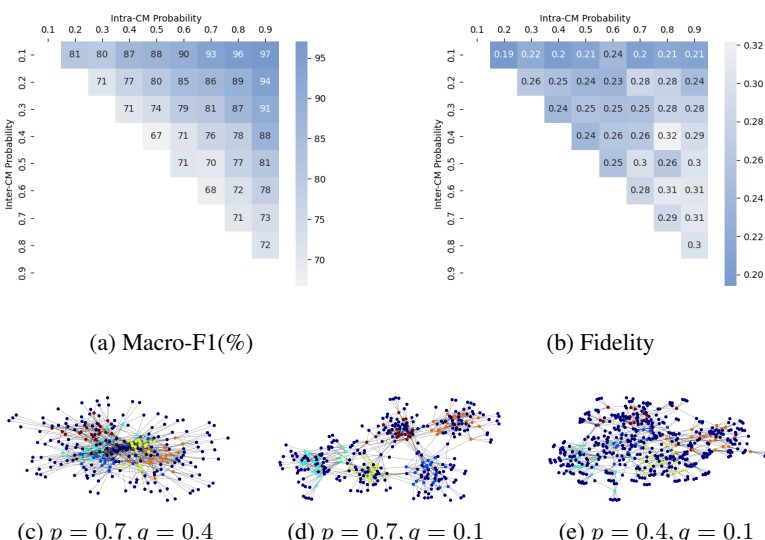

(a) Macro-F1(%)                    (b) Fidelity

(c) $p = 0.7, q = 0.4$        (d) $p = 0.7, q = 0.1$        (e) $p = 0.4, q = 0.1$

Figure 5: Visualization of Synthetic IMDB in Different Intra-/Inter-Cluster Probabilities. Dark blue represents minor nodes, and other colors indicate target nodes on different labels.

## 4.2 EVALUATION METRICS

We repeated all experiments five times and evaluated performance using average Micro-F1 and Macro-F1 for node prediction (Wang et al., 2019; Lv et al., 2021; Hong et al., 2023) and Fidelity for interpretation evaluation (Yuan et al., 2021; Li et al., 2022). See more details in the Appendix A.3.

## 5 RESULTS AND DISCUSSION

### 5.1 CLUSTER EXCLUSION CAN BE ADJUSTED FLEXIBLY TO AFFECT MODEL PERFORMANCE

Intra-CM probability $p$ and Inter-CM probability $q$ determine the degree of cluster exclusion of the synthetic HINs. The higher $p$ and the lower $q$ increases cluster exclusion, which produces a purer and better-organized graph structure. Due to the flexibility of our proposed framework, this degree of exclusion can be flexibly adjusted. To evaluate how such an adjustment alters model performance, we benchmark the performance of the HGT on the synthetic IMDB (Syn-IMDB) with different $p, q$ with $p > q$. As illustrated in Fig. 5a, the HGT model performs better in terms of Macro-F1 as the $p$ increases, as Syn-IMDB has more tightly connected clusters or a higher cluster exclusion. Conversely, a decrease in $q$ introduces more noise, leading to a lower cluster exclusion and worse performance. We also visualize the synthetic IMDB graphs with three settings, shown in Fig. 5c, 5d, 5e. The above results demonstrated that we can use $p$ and $q$ to control the exclusion of clusters within the generating synthetic HIN and benchmark the ability of HGNN graph learning. This allows us to control graph generation with high flexibility. Similar trends can also be observed for SimpleHGN and TreeXGNN; details can be found in Fig. 9 in the Appendix.

### 5.2 MAJOR MOTIFS ACHIEVE LOW FIDELITY

Fig. 5b illustrates the variations in fidelity for HGT across different degrees of cluster exclusion. The megatrend is similar to HGT's Macro-F1 score, with the purer (higher degree of cluster exclusions) synthetic HINs fidelity score being better (lower). Notably, the fidelity remains stable regardless of changes in the Intra-CM probability $p$. While higher Intra-CM probabilities may introduce additional information beyond the major motifs, the fidelity of the model does not undergo significant changes, shown in Fig. 6a. This finding confirms that the major motif designed indeed serves as the primary cause for accurate predictions of GNN models. Fig. 6b clearly shows that the fidelity

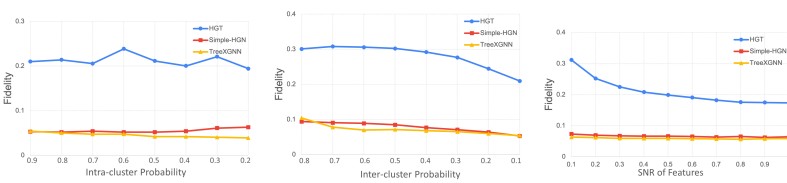

(a) Intra-Cluster Probability (b) Inter-Cluster Probability (c) SNR of Node Features

Figure 6: Fidelity of Major Motifs under Different SynHING Parameter Settings

Table 1: Ablation studies of SynHING: We replace the critical modules of SynHING with random functions to verify the importance. The Random-motifs turn off the MMG module. The Random-Merge entails randomly merging nodes without conducting Intra-/Inter-CM.

|  | SynIMDB | | Random-Motifs | | Random-Merge | |
| --- | --- | --- | --- | --- | --- | --- |
|  | Macro-F1 (%) | Micro-F1 (%) | Macro-F1 (%) | Micro-F1 (%) | Macro-F1 (%) | Micro-F1 (%) |
| HAN | $82.37 \pm 0.45$ | $82.42 \pm 0.52$ | $77.21 \pm 0.69$ | $77.20 \pm 0.81$ | $37.31 \pm 5.57$ | $41.47 \pm 4.40$ |
| HGT | $87.86 \pm 0.30$ | $87.88 \pm 0.31$ | $80.99 \pm 0.55$ | $80.97 \pm 0.58$ | $68.52 \pm 3.74$ | $69.11 \pm 3.24$ |
| SimpleHGN | $87.60 \pm 0.49$ | $87.66 \pm 0.49$ | $83.04 \pm 0.48$ | $83.02 \pm 0.48$ | $72.77 \pm 6.33$ | $72.88 \pm 6.30$ |
| TreeXGNN | $87.68 \pm 0.35$ | $87.70 \pm 0.36$ | $81.77 \pm 1.14$ | $81.73 \pm 1.15$ | $68.92 \pm 6.57$ | $69.07 \pm 6.56$ |

decreases as the Inter-CM probability $q$ decreases while maintaining a fixed Intra-CM probability $p$. This decrease can be attributed to the increase in the exclusion of clusters, which causes models to primarily learn from the corresponding major motifs without interference from non-informative nodes and edges. Fig. 6c illustrates the relationship between fidelity and the SNR of node features. As the SNR increases, the amount of information associated with all node features also increases. Consequently, when the graph structure and major motif design remain unchanged, the models incorporate more node feature information, thereby improving fidelity. The above experimental results can confirm that the design of major motifs can indeed represent the most crucial graph patterns and can serve as effective ground-truth explanations.

## 5.3 ABLATION STUDIES OF SYNHING

To evaluate the impact of each module in the SynHING framework, we performed ablation studies by removing critical modules one at a time on the IMDB dataset. More specifically, the Random-Motifs study turns off the MMG module and randomly generated motifs. The Random-Merge study entails randomly merging nodes without conducting Intra-Cluster and Inter-Cluster Merges. Table 1 shows that compared to the original SynIMDB, the performance of Random-Motifs decreases significantly on all HGNNs, HAN, HGT, SimpleHGN, and TreeXGNN (-5.16%, -6.87%, -4.56%, and -5.91% in Macro-F1, respectively). This drop highlights the importance of the MMG module. Furthermore, the performance of Random-Merge declines even further (-45.06%, -19.34%, -14.83%, and -18.76% in Macro-F1 for HAN, HGT, SimpleHGN, and TreeXGNN, respectively), revealing consistent trends across these HGNNs, demonstrating the effectiveness of the proposed Merge method for generating synthetic HINs.

## 5.4 HOW SIMILAR IS THE SYNTHETIC GRAPH TO THE ACTUAL GRAPH?

To answer this question, we assess high-level similarity by pretraining on the synthetic graph and finetuning on the responding actual graph, inspired by the fact that without careful selection of pretraining tasks, the transfer of knowledge between diverse semantics can lead to negative transfer (Hu et al., 2020a; Rosenstein et al., 2005). Specifically, we evaluate similarity from two perspectives: (i) If the synthetic and reference graphs are similar, we expect to see a positive transfer. (ii) If we maliciously destroy the structure and features of the synthetic graphs, a negative transfer would occur, which could decrease performance. See more implementation details in Appendix A.8.

For the positive transfer experiment, we compare the performance of finetuning the model pretrained on the synthetic graph with and without pertaining. The results, shown in Table 2, demonstrated pretraining on synthetic HINs significantly improved performance in the four datasets. In the IMDB dataset, we increased Macro-F1 by up to 3% with HGT and 2% with SimpleHGN. On

IMDB, Recipe, and ACM, the standard deviation was significantly reduced, enhancing the model's stability and slightly increasing the performance. The consistent trends across two HGNNs on four datasets demonstrate solid positive transfer effectiveness.

Table 2: Performance comparison of HGT and SimpleHGN: With and without pretraining on synthetic HINs and fine-tuning on real-world graphs. We use boldface to highlight performance improvements.

| Dataset | Pretrained on | HGT | | SimpleHGN | |
|---|---|---|---|---|---|
| | | Macro-F1 | Micro-F1 | Macro-F1 | Micro-F1 |
| IMDB | - | $63.00 \pm 1.19$ | $67.20 \pm 0.57$ | $63.53 \pm 1.36$ | $67.36 \pm 0.57$ |
| | **Syn-IMDB** | **66.10 $\pm$ 0.21** | **68.03 $\pm$ 0.53** | **65.52 $\pm$ 0.50** | **68.45 $\pm$ 0.53** |
| Recipe | - | $57.26 \pm 1.84$ | $56.98 \pm 2.02$ | $60.29 \pm 1.31$ | $60.15 \pm 1.41$ |
| | **Syn-Recipe** | **57.82 $\pm$ 0.46** | **57.83 $\pm$ 0.64** | **60.40 $\pm$ 0.22** | **60.21 $\pm$ 0.23** |
| ACM | - | $91.12 \pm 0.76$ | $91.00 \pm 0.76$ | $93.42 \pm 0.44$ | $93.35 \pm 0.45$ |
| | **Syn-ACM** | **92.55 $\pm$ 0.20** | **92.54 $\pm$ 0.21** | **94.16 $\pm$ 0.43** | **94.11 $\pm$ 0.44** |
| DBLP | - | $93.01 \pm 0.23$ | $93.49 \pm 0.25$ | $94.01 \pm 0.24$ | $94.46 \pm 0.22$ |
| | **Syn-DBLP** | **93.88 $\pm$ 0.25** | **94.35 $\pm$ 0.23** | **94.27 $\pm$ 0.58** | **94.73 $\pm$ 0.56** |

For the negative transfer experiment, malicious graphs are created by: (i) Node shuffling involves row-wise shuffling the adjacency matrix $A$ corresponding to the graph, breaking the homophily of the synthetic graph. (ii) Feature shuffling involves row-wise shuffling the feature matrix $F$. Table 3 shows that malicious synthetic graphs obviously cause negative transfer. In most cases, the feature shuffling has a greater impact. It is speculated that the mismatch between the node features and the labels causes more damage to the overall message passing. Noted, when the users focus on explainable ground truths, the generated synthetic graphs do not need to resemble the reference graphs. SynHING can freely generate brand-new HINs through user-defined major motifs.

Table 3: Performance comparison of HGT: Pretraining on node shuffled and feature shuffled synthetic HINs and finetuning on real HINs. We use boldface to highlight the lowest score and an underline to indicate the second-lowest.

| | Pretrain on SynHING | Macro-F1 | Micro-F1 |
|---|---|---|---|
| IMDB | w/o Shuffled | $66.10 \pm 0.21$ | $68.03 \pm 0.53$ |
| | Node Shuffled | $\underline{64.54 \pm 0.58}$ | $\underline{67.44 \pm 0.59}$ |
| | **Feature Shuffled** | **62.06 $\pm$ 1.28** | **63.96 $\pm$ 0.79** |
| Recipe | w/o Shuffled | $57.82 \pm 0.46$ | $57.83 \pm 0.64$ |
| | **Node Shuffled** | **47.87 $\pm$ 0.83** | **47.66 $\pm$ 0.88** |
| | Feature Shuffled | $\underline{55.46 \pm 1.09}$ | $\underline{55.55 \pm 1.11}$ |
| ACM | w/o Shuffled | $92.55 \pm 0.20$ | $92.54 \pm 0.21$ |
| | Node Shuffled | $\underline{90.45 \pm 0.49}$ | $\underline{90.45 \pm 0.48}$ |
| | **Feature Shuffled** | **89.02 $\pm$ 1.54** | **89.09 $\pm$ 1.46** |
| DBLP | w/o Shuffled | $93.88 \pm 0.25$ | $94.35 \pm 0.23$ |
| | Node Shuffled | $\underline{93.56 \pm 0.32}$ | $\underline{94.06 \pm 0.30}$ |
| | **Feature Shuffled** | **93.25 $\pm$ 0.29** | **93.75 $\pm$ 0.30** |

We also applied a statistical method, Comparing Degree Distribution (CDD) (Darabi et al., 2023), measuring the structure similarity between real and synthetic HINs. The results indicate that SynHING effectively regulates the generation of synthetic HINs. See more details in Appendix A.7.

## 6 CONCLUSION

We present SynHING, a novel method for generating synthetic HINs, leveraging the real-world HINs as references, systematically generating the major motifs for explanations, and using Intra-/Inter-Cluster Merges to merge multiple groups of base subgraphs to generate synthetic HINs of any specified sizes. SynHING has been validated using four datasets covering three distinct application categories, demonstrating its generality and practicality. we address the scarcity of heterogeneous graph datasets and overcome the need for such datasets in the domain of GNN explanations. To the best of our knowledge, our work introduces the first framework for generating synthetic heterogeneous graphs with ground-truth explanations. Additionally, we design a comprehensive framework for generating diverse synthetic HINs that can be flexibly adjusted and provide a solid foundation for future research on heterogeneous GNN explanations.

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

## A APPENDIX

### A.1 SYNHING'S COMPLEXITY AND SCALABILITY

In this section, we theoretically analyze the complexity of the SynHING framework module by module to demonstrate its scalability. Let $N$ represent the motif number, determining the scale of the generated graph. We demonstrate that the total time complexity for SynHING is $O(N)$. For simplicity, we omit node type in the analysis for both Intra-CM and Inter-CM, as nodes merge only with those of the same type, making the complexity linear to the number of types. The generations of the motif (MMG) and the base subgraph (BSG) can be parallelized, and execution time is linear to the number of items. Therefore, the time complexity of MMG and BSG is $O(N)$.

The complexity of Intra-CM is analyzed step-by-step as follows:

 (i) Eq.(3), we determine $n_{\text{intra}}$, the number pairs to be sampled.

 (ii) Eq.(4), we sample $n_{\text{intra}}$ nodes from $\mathcal{V}_y$ and $\mathcal{V}_i$, and pairing them as $\mathfrak{P}_{\text{intra}}$.

 (iii) Merge process eq.(5).

Table 4: Statistics of three heterogeneous graph datasets

| | #Nodes | #Node Types | #Edges | #Edge Types | Target Node | | #Classes |
| | | | | | Type | #Features | |
| --- | --- | --- | --- | --- | --- | --- | --- |
| IMDB | 19,933 | 4 | 80,682 | 6 | Movie | 3,489 | 5 |
| Recipe | 53,428 | 3 | 1,049,048 | 4 | Recipe | 1 | 5 |
| ACM | 10,967 | 4 | 551,970 | 8 | Paper | 1,902 | 3 |
| DBLP | 27,303 | 4 | 296,492 | 6 | Author | 340 | 4 |

(iv) We offset the "incoming" subgraph $S_i$ by the maximum IDs of $C_y$ (graph disjoint union).

(v) Drop the selected nodes in $\mathcal{V}_i$.

(vi) Reindex the edges in $\mathcal{E}_i$ based on the mapping determined by $\mathfrak{P}_{\text{intra}}$.

The complexity (i), (ii), and (v) are $O(|\mathcal{V}_i|)$. The complexity of steps (iv) is $O(|\mathcal{V}_i| + |\mathcal{E}_i|)$. The complexity of step (vi) is $O(|\mathcal{E}_i|)$. One iteration complexity is $O(|\mathcal{V}_i| + |\mathcal{E}_i|)$. There will be $(N - |\mathcal{Y}|)$ iterations, making the total complexity $O(N|\mathcal{V}_i| + N|\mathcal{E}_i|)$ or $O(N)$, as $|\mathcal{V}_i|$ and $|\mathcal{E}_i|$ are the number nodes and edges in the base subgraph, which are constant w.r.t. $N$.

Following a similar process, the complexity of Inter-CM is analyzed:

(i) Eq.(6) and Eq.(7), we identify all $\binom{|\mathcal{Y}|}{2}$ combinations of clusters and determine the number of pairs that need to be merged for each combination.

(ii) After the pair number has been determined, we derive the node number that needs to be merged for each cluster. We randomly select nodes from each cluster based on this number without replacement. (iii) Merge process eq.(8).

(iii) We offset all the clusters $C_y$. (the graph disjoint union in eq.(8)).

(iv) Drop one of the nodes in each pair in $\mathfrak{P}_{\text{inter}}$ in $\mathcal{V}_y$ for each cluster.

(v) Reindex the edges in $\mathcal{E}_y$ for each cluster based on the mapping determined by $\mathfrak{P}_{\text{inter}}$.

The complexity of (iv), (v), and (vi) are $O(\sum_{y \in \mathcal{Y}}(|\mathcal{V}_y| + |E_y|))$, $O(\sum_{y \in \mathcal{Y}} |\mathcal{V}_y|)$, and $O(\sum_{\mathcal{Y}} |\mathcal{E}_y|)$. Since $\sum_{y \in \mathcal{Y}} |\mathcal{V}_y| \leq N|\mathcal{V}_i|, \sum_{y \in \mathcal{Y}} |\mathcal{E}_y| \leq N|\mathcal{E}_i|$. The complexity of Inter-CM is $O(N|\mathcal{V}_i| + N|\mathcal{E}_i|) = O(N)$.

Overall, SynHING can generate large-scale HINs in a reasonable timeframe, with the graph scale determined by motif number $N$ and complexity of $O(N)$.

## A.2 DATASETS AND HGNNS

To evaluate the SynHING framework, we generate synthetic graphs based on four well-known HIN node classification datasets: IMDB 1, Recipe (Majumder et al., 2019), ACM (Wang et al., 2019), and DBLP 2. The IMDB dataset is a collection of movie data that requires predicting the various genres associated with each movie and following the common setting as previous papers (Lv et al., 2021; Hong et al., 2023). The Recipe dataset is gathered from Food.com and includes food recipes and user-recipe interactions. We excluded recipes with fewer than three steps, those with fewer than four or more than 20 ingredients, and users with fewer than four reviews. We select recipes as the target node and identify techniques used in recipes as labels. Then, we choose five techniques to create the recipe graph. On the other hand, ACM and DBLP are citation networks with different goals. ACM aims to predict paper labels, while DBLP focuses on predicting author labels. Fig. 7 illustrates the graph schema of the four heterogeneous graph datasets.

Table 4 presents the statistics of the four datasets, including the number of node and edge types, the number of nodes and edges, the number of target node features, and the number of classes.

## A.3 EVALUATION METRICS

We use Micro-F1 and Macro-F1 as evaluation metrics for node classification and fidelity for explanation evaluation. Micro-F1 scoring assesses a model's predictions across all samples, with a tendency to emphasize the majority category. In contrast, Macro-F1 scoring equally weights each

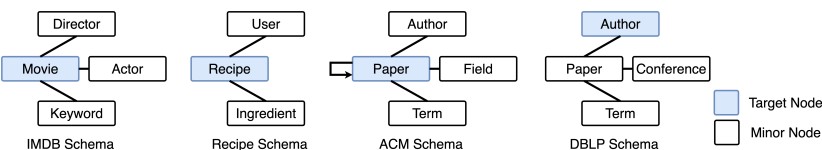

Figure 7: Graph schema of the four heterogeneous graph datasets

category, promoting a balanced evaluation of data across different categories. Therefore, we mainly use Macro-F1 as the major evaluation metric (Wang et al., 2019; Lv et al., 2021; Hong et al., 2023).

Fidelity is a metric commonly used to evaluate the performance of the explanation model (Yuan et al., 2021; Li et al., 2022). It measures how closely related the explanations are to the model's predictions. If the critical information is included in the explanation subgraph, the classification model prediction probability should be close to the original prediction, resulting in low fidelity. We use fidelity as the evaluation metric to support that the major motifs can be excellent explanations of ground truths. The following are the details of the fidelity score:

$$Fidelity = \frac{1}{N} \sum_{i=1}^{N} \frac{1}{L} \sum_{l=1}^{L} \left\| f(G_i)_{y_l} - f(\hat{G}_i)_{y_l} \right\|, \tag{9}$$

where $f(G_i)_{y_l}$ and $f(\hat{G}_i)_{y_l}$ denote the prediction probability of $y_l$ of the original graph $G_i$ and major motifs $\hat{G}_i$ (explanation subgraph), respectively. We denote $N$ as the total number of target node samples and $L$ as the number of node labels.

### A.4 BENCHMARK HETEROGENEOUS GRAPH NEURAL NETWORKS

We used three different concept HGNN models to validate the synthetic graphs. Model parameters follow paper recommendations. The following briefly introduces the models: (1) HGT (Hu et al., 2020b) adopts a transformer-based design for handling different node and edge types without manually defining the meta-path for the HGNN model. (2) SimpleHGN (Lv et al., 2021) introduces the attention mechanism, projects different node-type features to the shared feature space, and then uses GAT as the HGNN backbone. (3) TreeXGNN (Hong et al., 2023) leverages the decision tree-based model XGBoost to enhance the node feature extraction, assisting the HGNN model in getting more prosperous and meaningful information.

As the SynHING framework is modulized and can be highly customized according to the characteristics of the reference datasets, we conduct a series of experiments to examine the influence of different tunable parameters on the synthetic HINs. In order to evaluate the performance of SynHING, we utilize the transductive learning approach for node classification tasks and randomly select 24% of the target nodes for training, 6% for validation, and 70% for testing (Wang et al., 2019; Lv et al., 2021; Hong et al., 2023). We repeated all experiments five times and evaluated performance using average Micro-F1 and Macro-F1 for node prediction and fidelity for interpretation evaluation.

### A.5 SYNTHETIC HINS WITH GROUND-TRUTH EXPLANATIONS

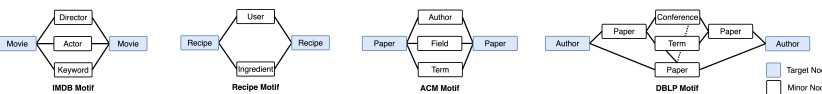

Figure 8: Major Motifs of the Four Heterogeneous Graph Datasets

We evaluate SynHING using three HGNNs on four synthetic HINs (with Syn- in front) based on their corresponding real-world graphs, shown in Table 5. HGNNs achieve better performance on Macro-F1 and Micro-F1 scores for learning and inferencing on synthetic graphs compared to real graphs. These improvements can be attributed to the designated major motifs in synthetic graphs, shown in Fig. 8, which provide ground-truth explanations for assessing explainability methods and result in synthetic graphs containing purer information for graph learning. We mimic the graph

Table 5: Performance comparison of three HGNNs on real and synthetic HINs

| | IMDB | | Recipe | | ACM | | DBLP | |
|---|---|---|---|---|---|---|---|---|
| | Macro-F1 (%) | Micro-F1 (%) | Macro-F1 (%) | Micro-F1 (%) | Macro-F1 (%) | Micro-F1 (%) | Macro-F1 (%) | Micro-F1 (%) |
| HGT | 63.00 ± 1.19 | 67.20 ± 0.57 | 57.26 ± 1.84 | 56.98 ± 2.02 | 91.12 ± 0.76 | 91.00 ± 0.76 | 93.01 ± 0.23 | 93.49 ± 0.25 |
| SimpleHGN | 63.53 ± 1.36 | 67.36 ± 0.57 | 60.29 ± 1.31 | 60.15 ± 1.41 | 93.42 ± 0.44 | 93.35 ± 0.45 | 94.01 ± 0.24 | 94.46 ± 0.22 |
| TreeXGNN | 65.59 ± 0.89 | 69.28 ± 0.64 | 59.99 ± 0.94 | 59.97 ± 0.96 | 94.32 ± 0.54 | 94.29 ± 0.54 | 94.94 ± 0.63 | 95.24 ± 0.59 |
| | SynIMDB | | SynRecipe | | SynACM | | SynDBLP | |
| | Macro-F1 (%) | Micro-F1 (%) | Macro-F1 (%) | Micro-F1 (%) | Macro-F1 (%) | Micro-F1 (%) | Macro-F1 (%) | Micro-F1 (%) |
| HGT | 87.86 ± 0.30 | 87.88 ± 0.31 | 87.95 ± 1.90 | 87.95 ± 1.88 | 99.45 ± 0.30 | 99.46 ± 0.30 | 97.97 ± 0.75 | 97.99 ± 0.74 |
| SimpleHGN | 87.60 ± 0.49 | 87.66 ± 0.49 | 87.82 ± 0.23 | 87.83 ± 0.23 | 99.41 ± 0.37 | 99.41 ± 0.37 | 98.48 ± 0.41 | 98.48 ± 0.41 |
| TreeXGNN | 87.68 ± 0.35 | 87.70 ± 0.36 | 86.68 ± 0.37 | 86.72 ± 0.36 | 99.12 ± 0.67 | 99.12 ± 0.66 | 99.18 ± 0.18 | 99.18 ± 0.18 |

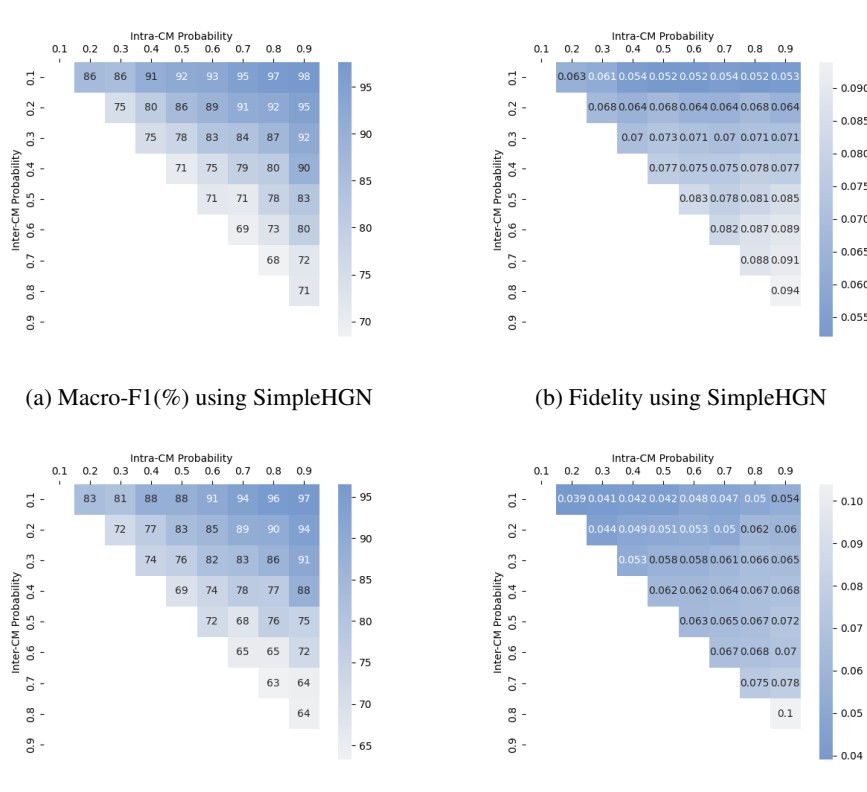

(a) Macro-F1(%) using SimpleHGN

(b) Fidelity using SimpleHGN

(c) Macro-F1 using TreeXGNN(%)

(d) Fidelity using TreeXGNN

Figure 9: Macro-F1 and Fidelity of Synthetic IMDB in Different Intra-/Inter-Cluster Probabilities across Different HGNNs

properties of the reference graph and identify the parameters for generating the synthetic graph. This selection ensures that the resulting synthetic graphs closely approximate the graph structure of the referenced graphs. In addition, the degree of exclusion in SynHING can be customized for different motifs and datasets, which will be discussed in the next subsection.

## A.6 MORE EXPERIMENTAL RESULTS

Fig. 10a illustrates the performance changes of HGT, Simple-HGN, and TreeXGNN at different SNRs of the features. It shows that as the SNR increases, the disparity between node features in different groups widens, and it is easier to discriminate different clusters only based on their features. Consequently, when the classification model makes predictions, it can leverage this additional information in the nodes, leading to improved performance in classification tasks.

We also explored the impact of adjusting the number of major motifs shown in Fig. 10b, which directly affects the number of target nodes and the size of the synthetic graph dataset. It is important to note that since we kept the hyperparameter settings of the classification model consistent with the

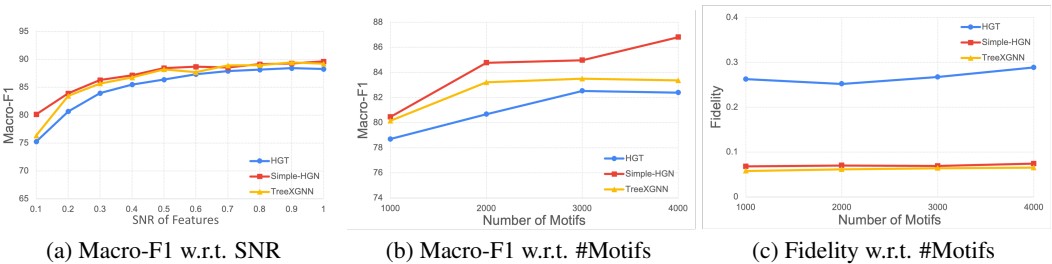

(a) Macro-F1 w.r.t. SNR     (b) Macro-F1 w.r.t. #Motifs     (c) Fidelity w.r.t. #Motifs

Figure 10: Macro-F1 and Fidelity of Synthetic IMDB in Different SNR and Number of Motifs

original values, rather than fine-tuning them for each synthetic graph dataset, reducing the dataset size to half caused the model to become overfitted, resulting in a decline in performance.

The fidelity results of HGT, Simple-HGN, and TreeXGNN for varying numbers of major motifs are shown in Fig. 10c. When adjusting the number of motifs, which corresponds to the size of the graph, the fidelity performance remains stable.

## A.7 Approximating Referenced Graph

Users can customize the synthetic graph for various scenarios using the parameters of SynHING, including the number of major motifs $N$, the number of clusters $|\mathcal{Y}|$, the Intra-CM probabilities $p^\phi$, the Inter-CM probabilities $q^\phi$, and the signal-to-noise ratio (SNR) of features $\alpha/\beta$. For example, adjusting Intra-CM probability $p^\phi$ and Inter-CM probability $q^\phi$ results in changes in the exclusion of clusters and difficulty of the synthetic graph. However, these parameters can also be directly determined by the referenced graph $\hat{G}$. Although some statistical properties and network schema have been used for generating graphs, it's further demonstrated that the synthetic graph can approximate the referenced graph more closely by adjusting these parameters. The number of major motifs $N$ can be set as half of the number of target nodes in $\hat{G}$, i.e. $N = \frac{1}{2}|\hat{\mathcal{V}}^{\phi_0}|$, since each motif contains exactly 2 target nodes. The number of clusters can be determined by the number of labels $|\hat{\mathcal{Y}}|$ in $\hat{G}$. The SNR of features $\alpha/\beta$ can adjust the difficulty of the task on $\tilde{G}$, or users can determine the means and variances of clusters of features by maximum likelihood estimation.

The Intra-/Inter-CM probabilities $p^\phi, q^\phi$ for minor node type $\phi \neq \phi_0$ control the exclusion of clusters, the degree distributions of source nodes, and their counts in the resulting graph $\tilde{G}$. For instance, in Fig. 11, we observe the node degree distributions for minor node types in both real-world IMDB and SynIMDB, with $p = 0.7$ and $q = 0.3$. In contrast, Fig. 12 compares these distributions with SynIMDB using different probabilities: $p = 0.9, q = 0.8$, and $p = 0.2, q = 0.1$. As depicted, improper selection of $p$ and $q$ can lead to notable deviations in the degree distribution of minor node types.

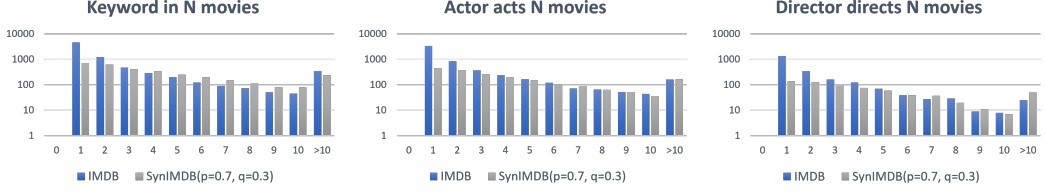

Figure 11: Degree Distributions of Minor Node Types in IMDB and SynIMDB

We further applied a statistical-based method, Comparing Degree Distribution (CDD) (Darabi et al., 2023), to measure the structure similarity between real and synthetic graphs. The CDD value ranges between 0 and 1, with 1 indicating that the distribution of the two structures is exactly the same. We applied the settings as Fig. 11 and Fig. 12 for structure similarity analysis. Table 6 indicates that the generated SynIMDB can be controlled by the Intra-CM/Inter-CM ratio that influences the

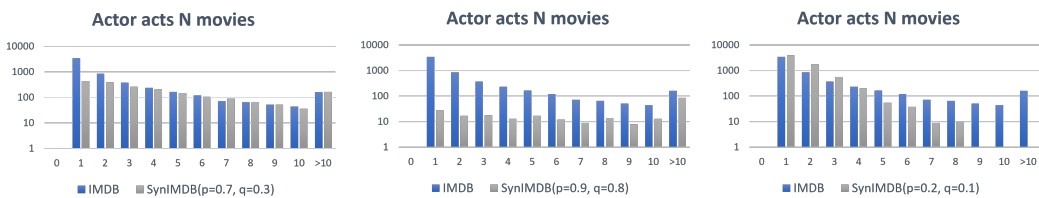

Figure 12: Comparison of Degree Distribution Deviations in SynIMDB with Varying Intra-/Inter-CM Probabilities

similarity with real IMDB. When p=0.7 and q=0.3, Macro-CDD and Micro-CDD are 0.8545 and 0.8279, respectively, which is the most similar to the real IMDB compared to the other two settings. This result highlights the effectiveness of SynHING in regulating the generation of synthetic HINs.

Table 6: Comparing Degree Distribution (CDD) between IMDB and SynIMDB with Varying Intra-/Inter-CM Probabilities

| Intra-CM (p), Inter-CM (q) | Macro-CDD | Micro-CDD |
|---|---|---|
| p=0.7, q=0.3 | 0.8545 | 0.8279 |
| p=0.9, q=0.8 | 0.7636 | 0.7612 |
| p=0.2, q=0.1 | 0.7597 | 0.7354 |

## A.8 PRETRAINING AND FINETUNING

In this study, we employ synthetic graphs for pretraining. Models are pretrained based on the recommended settings from their respective original papers, with early stopping applied after 30 epochs without validation set improvement. For finetuning, the weights of the HGNN backbone, excluding the adapter layer that maps the heterogeneous features into shared space, are inherited from the pre-trained model. We note that the weights of the backbone and adapter are trained using different learning rates, as the results are sensitive to the learning rate. For instance, while finetuning from pretrained weights, a lower learning rate for the backbone and a higher learning rate for the adapter generally yield better results, whereas a higher learning rate for the backbone and a lower learning rate for the adapter generally leads to better performance when learning from scratch. Consequently, we conduct a grid search for learning rates in both scenarios, as presented in Tables 2 and 3. For the learning rate of the backbone, we try values of $\{10^{-3}, 10^{-4}\}$. For that of the adapter, we try values of $\{1, 5\} \times \{10^{-2}, 10^{-3}, 10^{-4}\}$.

## A.9 IMPLEMENTED ON HGNN EXPLAINER

For our initial testing, we utilized synthetic ACM and DBLP datasets and inputted them into the xPath framework (Li et al., 2023b). The synthetic IMDB dataset we generated is a multiple-choice dataset. Since xPath does not support this, we skipped it for now. We utilized xPath's default parameters, including the HGNN encoder and explainer. We followed the instructions in xPath, which involved two main steps: (1) Training the HGNN and (2) Generating explanations.

We used HGT as our backbone prediction model. During the training stage, it can effectively converge and achieve solid performance (Macro-F1=99.42%, Micro-F1=99.43%) and (Macro-F1=80.06%, Micro-F1=80.81%) on SynACM and SynDBLP, respectively. During the explanation stage, xPath can successfully generate an explanations subgraph with decent accuracy fidelity and probability fidelity, Facc and Fprob (Yuan et al., 2020): Facc=0.15665, Fprob=0.15297 on SynACM and Facc=0.16935, Fprob=0.08701 on SynDBLP, both presenting quite reasonable scores. The above preliminary results show that our generated synthetic datasets can indeed be used to evaluate HGNN explanation algorithms. This warrants a more complete further exploration in future work.

## A.10 COMPUTING RESOURCES

In our experiments, GNN learning utilized an NVIDIA RTX 3060, with fitting a GNN on a heterogeneous information network (HIN) taking under an hour. Graph generation algorithms were executed on a CPU, with each graph requiring less than an hour to generate.

