# OpenReview forum: "SynHING: Synthetic Heterogeneous Information Network Generation for Graph Learning and Explanation"
_ICLR.cc/2025/Conference — Submitted to ICLR 2025_

### Official Review · Reviewer_kxr8 · 2024-10-29

**Soundness:** 3
**Presentation:** 3
**Contribution:** 3
**Rating:** 6
**Confidence:** 3

**Summary:**

This paper focuses on the scarcity of heterogeneous graph datasets and overcomes the need for such datasets in the domain of GNN explanations. In particular, it proposes a novel SynHING framework for generating synthetic HINs. This method leverages the real-world HINs as references and systematically generates the major motifs for explanations. Extensive experiments have demonstrated its generality and practicality.

**Strengths:**

1. Originality: This paper introduces the first framework for generating synthetic heterogeneous graphs with ground-truth explanations.

2. Clarity: The paper's writing in the methodology section is logically clear.  The reason behind each step of the method design is also explained in detail.

3. Significance: The framework for generating diverse synthetic HINs that can be flexibly adjusted will provide a solid foundation for future research on heterogeneous GNN explanations.

**Weaknesses:**

1. The size of many figures in this paper needs to be adjusted. For example, Figure 2 is too small, which is not conducive to reading.

2.  The writing of this paper could be improved. For instance, there is a significant gap between the first and second paragraphs of the introduction.

3.  The techniques used in this paper for generating synthetic HINs are relatively normal, focusing only on the degree of nodes in the graph to consider the relationship with real graphs.

4.  This paper only validates the methods on node-level tasks, and it is unclear whether this method can be transferred to graph-level explanation task scenarios.

5. The design of the paper's experimental section needs improvement. First, more tabular types of experiments should be used to conduct a more intuitive quantitative analysis. Second, important experimental results should be presented at the forefront.

**Questions:**

1. I would like to know if this method can assist with graph-level explanation tasks in heterogeneous graphs.

---

> ### Author Response · Authors · 2024-11-25
>
> We want to thank the reviewer for acknowledging our proposed framework's originality and for recognizing our methodology's quality and precise writing. We appreciate the reviewer's praise for the framework's flexibility and the significant impact of our research, which serves as a solid foundation for future research on HGNN explanations.
>
> [W1,2] We appreciate the reviewer's suggestion and revised the paper accordingly.
>
> [W3] In this work, we present SynHING, a novel method for generating synthetic HINs, leveraging the real-world HINs as references, systematically generating the major motifs for explanations, and using Intra-/Inter-CM to merge multiple groups of base subgraphs to generate synthetic HINs of any specified sizes. To the best of our knowledge, we are the first to introduce a synthetic HIN generation framework for graph learning and explanation.
>
> Existing research primarily focuses on homogeneous graphs, where synthetic graphs are created by randomly generating nodes and adding edges. However, this homogeneous approach is not suitable for HINs, as it can lead to the creation of illegal edges and offers limited control over the degree of nodes. The Merge methods can cleverly solve these issues and provide excellent generality (Section 3.5). We believe that SynHING's new button-up design is a significant technical advancement over traditional methods of randomly adding nodes, which can easily violate graph schema. SynHING enables the generation of synthetic HINs and allows for the creation of synthetic graphs with varying levels of exclusion by Inter-/Intra-CM tuning, which correspond to different noise levels.
> On the other hand, there are few community datasets that offer explanations for the ground truth of HGNN explainers. To address this issue, we extract major motifs from the reference graphs and generate ground truths to evaluate HGNN explainers (Section 3.3). As discussed in Section 2, previous works have mainly employed basic artificial motifs like houses and grids. These methods struggle to represent real-world motifs in HINs, making it difficult to adapt to diverse heterogeneous datasets and impractical for actual HIN applications.
>
> In summary, we tackle the shortage of HINs that mimic real-world graphs and overcome the need for datasets in the domain of GNN explanations. To the best of our knowledge, our work introduces the first framework for generating synthetic heterogeneous graphs with ground-truth explanations. Many modules and techniques we proposed are novel.
>
> [W4] Due to character limitations, please refer to 3db8 W4.
>
> [W5] We appreciate the reviewer's comments. We understand that in typical machine learning research, the main results of the experiment are compared with the state-of-the-art methods as the main table. However, what we propose in this paper is a method for generating synthetic datasets that can be used to evaluate different ML models. In our opinion, it cannot be presented like typical ML papers. Futhermore, there is no existing work generating synthetic HINs with explanation ground truths to compare with to the best of our knowledge. As a result, we carefully designed various experiments to prove the effectiveness of the proposed synthetic HIN generation algorithm from various aspects step by step. Below, we humbly explain the logical structure of our experimental section.
>
> In Section 5.1, we demonstrate the capability to generate synthetic HINs with varying levels of exclusion among clusters through Inter-/Intra-CM, reflecting different noise levels that previous methods were unable to achieve. Since SynHING can efficiently generate interpretable HINs by referring to major motifs, in Section 5.2, we demonstrate that the major motifs in the synthetic HINs can represent critical patterns in HINs and achieve excellent fidelity. Additionally, similar to other studies, we conducted ablation studies on the proposed method; Section 5.3 further assesses the impact of critical modules within SynHING.  As mentioned, we can generate synthetic HINs that resemble real HINs. In Section 5.4, we explore the similarities between them and derive a pretraining/fine-tuning approach to potentially enhance the HGNN prediction performance.
>
> We conduct comprehensive verification and analysis from various aspects. We hope that the organized experimental logic flow has effectively raised and addressed the essential research questions, emphasizing the research's significance and integrity.

---

> > ### Comment · Reviewer_kxr8 · 2024-11-26
> >
> > Thank you for your response. The author has addressed some of my concerns in the revised version, and I will raise my final score.

---

### Official Review · Reviewer_3db8 · 2024-10-30

**Soundness:** 3
**Presentation:** 3
**Contribution:** 4
**Rating:** 6
**Confidence:** 4

**Summary:**

This paper proposes a new method for synthesizing heterogeneous graphs called SynHING. SynHING generates synthetic heterogeneous graphs through modules such as Major Motif Generation, Base Subgraph Generation, Intra-Cluster Merge, Inter-Cluster Merge, and Node Feature Generation, allowing for flexible adjustment of the size of the HIN. Multiple experiments validate the effectiveness and scalability of the synthetic heterogeneous graphs.

**Strengths:**

1. This paper proposes a new direction for creating synthetic datasets for heterogeneous graphs, which is scarce in this area.
2. The methods in the paper are innovative, and each module is necessary and effective.
3. The experiments in the paper are sufficient and validate the effectiveness of the proposed synthetic graphs.

**Weaknesses:**

1. The images in the paper are too small and difficult to see clearly, especially Figure 2.
2. The authors should provide a detailed introduction of when synthetic graphs need to approximate reference graphs and when they should differ from them. In my view, synthetic graphs should address some of the shortcomings of the reference graphs; otherwise, Section 5.4 lacks significance.
3. I believe the authors need to conduct explainable experiments on the synthetic graphs to validate their effectiveness and to verify whether the ground truth is accurate. Potential models include: xPath[1] and HENCE-X[2].
4. The proposed synthetic dataset seems targeted at node classification tasks; can it also be applied to graph classification tasks?

Reference:
[1] Li, T., Deng, J., Shen, Y., Qiu, L., Yongxiang, H., & Cao, C. C. (2023, June). Towards fine-grained explainability for heterogeneous graph neural network. In Proceedings of the AAAI Conference on Artificial Intelligence (Vol. 37, No. 7, pp. 8640-8647).
[2] Lv, G., Zhang, C. J., & Chen, L. (2023). HENCE-X: Toward Heterogeneity-Agnostic Multi-Level Explainability for Deep Graph Networks. Proceedings of the VLDB Endowment, 16(11), 2990-3003.

**Questions:**

See Weaknesses.

---

> ### Author Response · Authors · 2024-11-25
>
> We appreciate the reviewer's recognition of the innovativeness and contribution of our research in generating synthetic heterogeneous graphs. We are also grateful for the acknowledgment of the flexibility of our framework, as well as the various experiments we conducted to validate the effectiveness and scalability of these synthetic heterogeneous graphs.
>
> [W1] We appreciate the reviewer's suggestion and adjusted the size of the figure as recommended.
>
> [W2] In SynHING, the degree of similarity between synthetic and real HINs can actually be adjusted according to the user's needs. We aim to provide a flexible and general synthetic heterogeneous information network generation framework, allowing users to easily generate various synthetic HIN baselines and equipping them with explanation ground truth for HGNN explainer research. We can determine the levels of exclusion between synthetic subgraphs by controlling the Intra-/Inter-Cluster Merge parameters, as mentioned in Section 3.5 and Figure 5. It can effectively generate synthetic HINs with different levels of exclusion to evaluate the performance of the HGNN encoders and explainers.
>
> To validate the effectiveness of the generated graphs, we pre-trained the HGNN model on synthetic graphs and fine-tuned it on real graphs to enhance predictive performance. We hypothesize that synthetic graphs that are similar to the reference graphs can address the limitations of real graphs and provide a suitable structure for HINs, serving as a strong starting point for fine-tuning. Experimental results show a positive transfer effect on real downstream tasks, as illustrated in Table 2, indicating exciting potential for further research.
> On the other hand, one of the main purposes of this work is to generate synthetic heterogeneous graphs with explainable ground truths. When the users focus on explainable ground truths, the reference graph is only treated as a starting point, and the generated synthetic graphs are not required to resemble the reference graphs. In this case, the user can freely determine the level of exclusions according to their needs.
> We included statements to clarify this in the revised manuscript.
>
> [W3] We appreciate the reviewer's deep interest in the synthetic HINs proposed in SynHING. The synthetic HINs we create aim to provide controllable datasets for fair use as benchmarks for HGNNs and HGNN explainers, not the other way around. Users can evaluate their explanation algorithms using our synthetic HINs, which are based on user-provided motifs as explainable ground truth. We provide a systematic method to generate HINs with an explanation ground truths for benchmark HGNN explainers, like previous works, BA-Shape, BA-Community, Tree-Cycles, and Tree-Grid, but in heterogeneous settings and mimic user-provided motifs. As much as we would like to provide more support to validate our approach, running explanation algorithms on our datasets does not serve the purpose. A good fidelity score from a certain algorithm only says it is a good explanation algorithm since it captures our pre-defined motif. We would like to do such a comprehensive analysis for future work. This is an important topic to be addressed but should be the follow-up step after we establish the validity of this paper.
>
>
> [W4] We appreciate the reviewer raising the question of graph classification. The answer is affirmative.  In previous studies [1, 2], the authors partition the community graph into small subgraphs for graph-level tasks, each associated with a unique label based on a major node within the subgraph. This major node can be replaced with the target node in our work, and our synthetic graph can be readily partitioned into the required subgraphs in a similar manner.
>
> Additionally, our proposed SynHING framework could potentially be adapted to generate graph-level HINs that involve stand-alone graphs (not subgraphs from a larger graph). This includes datasets such as MUTAG and NCI, which consist of molecular graphs. Specifically, the generation process can leverage pre-defined motifs, such as functional groups commonly found in molecules, as major motifs for the explanation. Further, random nodes are added to generate a new synthetic molecular graph that preserves key structural and functional characteristics. The construction process for such synthetic graphs requires further research to ensure their robustness and effectiveness for graph classification tasks.
>
>
> Ref:
>
> [1] Pinar Yanardag and SVN Vishwanathan. Deep graph kernels. In Proceedings of the 21th ACM SIGKDD international conference on knowledge discovery and data mining, pages 1365–1374, 2015.
>
> [2] Nouranizadeh, Amirhossein et al. “Maximum Entropy Weighted Independent Set Pooling for Graph Neural Networks.” ArXiv abs/2107.01410 (2021): n. pag.

---

> > ### Comment · Reviewer_3db8 · 2024-11-25
> > **Thank you.**
> >
> > Thank you for your response. One of the main purposes of this work is to generate synthetic heterogeneous graphs with explainable ground truths. Therefore, I still strongly believe that explainable experiments are necessary to enable the explainer to work on your dataset. Although we are unsure about the performance of the explainer on your dataset, this experiment is essential, as it is one of the motivations behind the introduction of your dataset.

---

> > > ### Author Response · Authors · 2024-11-27
> > >
> > > Thank you for your response. We conducted experiments based on your comment.
> > > Due to limited time and resources, we chose xPath, as the reviewer first mentioned, which is implemented in the DGL framework and published in AAAI with 7 citations.
> > > For our initial testing, we utilized synthetic ACM and DBLP datasets and inputted them into the xPath framework. The synthetic IMDB dataset we generated is a multiple-choice dataset. Since xPath does not support this, we skipped it for now. We utilized xPath's default parameters, including the HGNN encoder and explainer. We followed the instructions in xPath's README, which involved two main steps: (1) Training the HGNN and (2) Generating explanations.
> > >
> > > The experimental results of xPath are as follows :
> > >
> > > [ACM]
> > >
> > > - Training the HGNN:
> > >
> > >     "best_metrics": {
> > >         "epoch": 95.0,
> > >         "loss/train": 0.07013213634490967,
> > >         "macro_f1/train": 0.9943394599517282,
> > >         "micro_f1/train": 0.9944444444444445,
> > >         "loss/val": 0.032236095517873764,
> > >         "macro_f1/val": 1.0,
> > >         "micro_f1/val": 1.0,
> > >         "loss/test": 0.03415926173329353,
> > >         "macro_f1/test": 0.9942324751230855,
> > >         "micro_f1/test": 0.9942857142857143
> > >     }
> > >
> > > - Generating explanations:
> > >
> > >     Evaluating fidelity…
> > >
> > >     fmask acc: 0.15665, fmask prob: 0.15297
> > >
> > > [DBLP]
> > >
> > > - Training the HGNN:
> > >
> > >     "best_metrics": {
> > >         "epoch": 80.0,
> > >         "loss/train": 0.4520876407623291,
> > >         "macro_f1/train": 0.8522215296845654,
> > >         "micro_f1/train": 0.855087358684481,
> > >         "loss/val": 0.46361780166625977,
> > >         "macro_f1/val": 0.8401059872884211,
> > >         "micro_f1/val": 0.8442622950819673,
> > >         "loss/test": 0.5277144908905029,
> > >         "macro_f1/test": 0.800561688625627,
> > >         "micro_f1/test": 0.8080985915492958
> > >     }
> > >
> > > - Generating explanations:
> > >
> > >     Evaluating fidelity…
> > >
> > >     fmask acc: 0.16935, fmask prob: 0.08701
> > >
> > >
> > > The experimental results show that there are consistent phenomena in the two datasets.
> > > We used HGT as our backbone prediction model. During the training stage, it can effectively converge and achieve solid performance (Macro-F1=99.42%, Micro-F1=99.43%) and (Macro-F1=80.06%, Micro-F1=80.81%) on SynACM and SynDBLP, respectively. During the explanation stage, xPath can successfully generate an explanations subgraph with decent accuracy fidelity and probability fidelity, Facc and Fprob (Yuan et al.): Facc=0.15665, Fprob=0.15297 on SynACM and Facc=0.16935, Fprob=0.08701 on SynDBLP, both presenting quite reasonable scores.
> > >
> > > The above preliminary results show that our generated synthetic datasets can indeed be used to evaluate HGNN explanation algorithms. This warrants a more complete further exploration in future work. We included the above experimental results in the revised manuscript.
> > >
> > >
> > > Ref:
> > >
> > > Yuan, H.; Yu, H.; Gui, S.; and Ji, S. 2020. Explainability in graph neural networks: A taxonomic survey. arXiv preprint arXiv:2012.15445.

---

> > > > ### Comment · Reviewer_3db8 · 2024-11-29
> > > > **Thank you**
> > > >
> > > > Thank you for your reply. I will increase my score. I hope you can add more HGNN explainer examples in the revised manuscript in the future.

---

> > > > > ### Author Response · Authors · 2024-11-29
> > > > >
> > > > > Thanks for your positive feedback! We will definitely do that in the future.

---

### Official Review · Reviewer_4TeQ · 2024-11-01

**Soundness:** 2
**Presentation:** 2
**Contribution:** 3
**Rating:** 5
**Confidence:** 3

**Summary:**

The paper presents SynHING, a framework for generating synthetic heterogeneous information networks (HINs) designed to support graph neural network (GNN) learning and model interpretability. The framework synthesizes HINs by identifying and replicating core structural motifs from real-world graphs, creating diverse subgraphs, and merging them through controlled intra-cluster and inter-cluster methods to ensure realistic structure and statistical alignment with reference datasets.

**Strengths:**

-	The paper presents SynHING, which generates synthetic heterogeneous information networks specifically designed to support graph learning and GNN explainability.
-	SynHING’s modular design (e.g., motif generation, intra-cluster and inter-cluster merging) allows flexible control over the network structure, making it possible to model complex HIN characteristics.

**Weaknesses:**

-	While SynHING’s computational complexity is discussed, an empirical evaluation of scalability and runtime efficiency on large-scale datasets is not provided. The author mentions in line 339 that "the overall time complexity of SynHING is O(N)," which seems efficient. However, the largest graph size used in the experiments is only 53,428, which is insufficient to demonstrate the scalability of the method.
-	The purpose of SynHING is to generate HINs. To evaluate the effectiveness of SynHING, the author selects only three different HGNNs as baselines, which is insufficient to fully demonstrate the generalization capability of SynHING in generating HINs。
-	Evaluating how similar the synthetic graphs are to real graphs is a meaningful experiment. The author addresses this question through pre-training and fine-tuning, which seems reasonable. However, the author only reports the results for HGT. Reporting results for more baselines, with consistent performance trends, would be necessary to support the experimental conclusions。
-	The presentation of the paper has some formatting inconsistencies that require a thorough review: (1) The caption of Figure 5 ends with a period, while other figures do not. Normally, if a caption is a complete sentence, it should end with a period. (2) Different formats are used when referencing figures, such as "Fig. 3(a)," "Figure 4a," and "Fig. 5a.".

**Questions:**

See the weaknesses part.

---

> ### Author Response · Authors · 2024-11-25
>
> We appreciate the reviewer's acknowledgment of our research addressing the shortage of heterogeneous graph datasets and its support for graph learning and GNN explainability. Furthermore, we are grateful for the recognition of our framework's flexibility.
>
> [W1] We appreciate the reviewer for acknowledging that the theoretical time complexity of SynHING is O(N). In general, algorithms in computer science and data science that achieve O(N) complexity are considered well-designed. In this study, we implemented experiments on commonly available, typical-sized real-world open datasets. These datasets have been used in well-known studies, including HAN, MAGNN, GTN, SimpeHGN, and more. We believe they are representative of real-world scenarios. Due to limitations in time and computing resources during the rebuttal period, it is difficult for us to validate further on a larger dataset. We hope the reviewer can understand.
>
> [W2] We appreciate the reviewer for raising the question about the effectiveness of SynHING. To further illustrate this, we applied the generated synthetic HINs to an additional well-known meta-path-based HGNN encoder, HAN. The results are presented in revised Table 1.
>
> In the additional experiments, the performance of HAN aligns with trends observed in other models, further demonstrating the importance of SynHING modules. Overall, we evaluate synthetic HINs with four well-known category-based HGNNs: (1) Meta-path-based, HAN; (2) Transformer-based, HGT; (3) Shared feature space, SimpleHGN; and (4) Two-stage, TreeXGNN. The experimental results reveal consistent trends across these approaches, demonstrating the generalization capability of SynHING in generating HINs.
> We included additional experimental results in the revised manuscript.
>
> [W3] Thanks to the reviewer's suggestion, we have conducted the experiment of SimpleHGN pretraining/finetuning. The results are shown in revised Table 2.
>
> The table indicates that the experimental trends of SimpleHGN align with those of HGT. We perform pretraining/finetuning experiments on four datasets with two HGNN encoders. The comprehensive results demonstrate positive transfer effects across various datasets and HGNNs, confirming our experimental findings. We included additional SimpleHGN pretraining/finetuning results in the revised manuscript to support the experimental conclusions.
>
> [W4] We appreciate the reviewer's suggestion and revised the paper format accordingly.

---

> > ### Author Response · Authors · 2024-11-28
> >
> > To showcase the contribution of SynHING to advancing interpretable machine learning, we embarked on further experiments employing the HGNN explainer.
> >
> > For our initial tests, we input synthetic ACM and DBLP into the xPath framework [1], which is a GNN explainer specifically designed for HINs. In the training phase, we use HGT as the backbone prediction model, which can effectively converge and achieve solid performance (Macro-F1=99.42\%, Micro-F1=99.43\%) and (Macro-F1=80.06\%, Micro- F1=80.81\%) on SynACM and SynDBLP, respectively. During the explanation stage, xPath can successfully generate an explanations subgraph with decent accuracy fidelity and probability fidelity, Facc and Fprob \citep{Yuan2020ExplainabilityIG}: Facc=0.15665, Fprob=0.15297 on SynACM and Facc=0.16935, Fprob=0.08701 on SynDBLP, both presenting quite reasonable scores.
> >
> > The above results demonstrate that our synthetic heterogeneous information network generation framework indeed facilitates graph learning and interpretation.
> >
> > Ref:
> >
> > [1] Tong Li, Jiale Deng, Yanyan Shen, Luyu Qiu, Huang Yongxiang, and Caleb Chen Cao. Towards fine-grained explainability for heterogeneous graph neural network. Proceedings of the AAAI Conference on Artificial Intelligence, 37(7):8640–8647, Jun. 2023b. doi: 10.1609/aaai.v37i7.

---

> > > ### Author Response · Authors · 2024-12-04
> > >
> > > We have carefully reviewed and addressed the comments provided by the reviewers. We hope our revisions have successfully addressed your concerns and met your expectations.

---

### Official Review · Reviewer_7TVj · 2024-11-03

**Soundness:** 2
**Presentation:** 2
**Contribution:** 2
**Rating:** 5
**Confidence:** 4

**Summary:**

This paper introduces SynHING, the first framework designed for generating synthetic heterogeneous graphs with ground-truth explanations. SynHING designed to advance graph learning and explanation. The effectiveness of SynHING is validated using four datasets. This paper establishing a new benchmark for explainable

**Strengths:**

S1:  This paper is well-written, and the method is presented clearly.
S2: The article proposes a novel framework for generating synthetic heterogeneous information networks, which is an important contribution to the study of the interpretability and generalization capabilities of Graph Neural Networks
S3: The SynHING framework is capable of generating synthetic graphs with practical application backgrounds, such as community analysis and recommendation systems

**Weaknesses:**

S1: The document mentions some existing synthetic graph generation techniques, why not include them for comparison?
S2: In heterogeneous graphs, nodes typically have node types and node attributes. Why does the paper only generate attribute information for the target nodes?

**Questions:**

Seen Weaknesses

---

> ### Author Response · Authors · 2024-11-25
>
> We thank the reviewer for acknowledging the innovation of our proposed framework. We appreciate their recognition of the quality of our methodology, the clarity of our writing, and the significant contributions our work makes to the interpretability and generalization capabilities of GNN in practical application contexts.
>
> [S1] We appreciate the reviewer's question regarding the comparison with existing synthetic graph generation techniques. As much as we would like to compare with them, almost all focus on homogeneous rather than heterogeneous graphs. As a result, they could not apply to HINs as mentioned in Section 2.1 (Snijders & Nowicki, 1997) (Ying et al., 2019) (Albert & Baraba ́si, 2002) (Abbe, 2017) (Dwivedi et al., 2020) (Tsitsulin et al., 2020) (Rozemberczki et al., 2021).
>
> Specifically, traditional synthetic graph generation designs that focus on homogeneous graphs cannot be easily extended to HIN generation for the following reasons:
> 1. They focus on generating homogeneous graphs by randomly creating nodes and adding edges between them. If it were to be used to generate heterogeneous graphs, illegal edges would be generated that violate the existing graph schema.
> 2. They often use basic artificially structured motifs, such as houses and grids, and cannot reflect the real-world motifs in HINs.
> 3. They randomly add edges, which only increase node degree, without considering the overall graph structure in a meaningful way. In this case, it is difficult to match the node degree and structure well with the real-world HINs.
> 4. They fail to control exclusion between different clusters, which crucially affects the prediction accuracy on the node level in the follow-up tasks. This also reduces the explainability of the ground truth in the generated graph.
>
> As such, the limitations of the current homogeneous generation methods make it difficult to generate synthetic HINs. To address this issue, we designed Intra-/Inter-Merge Generation modules specifically tailored for HINs. These modules effectively resolve the illegal edge problem and help manage the node degree and exclusion levels among clusters.
>
> Moreover, there are no algorithms capable of generating HINs while also providing explanations of the ground truth. To tackle this gap, we propose using Major Motif Generation to create major motifs that are derived from and mimic real data. These motifs will serve as a ground truth for GNN explanation research. To verify the performance of the proposed SynHING, we compared it to randomly generated variants derived from conventional methods, as shown in Table 1 in Section 5.3. This comparison demonstrated the effectiveness of our proposed method. In the rebuttal period, we also added an additional HGNN and conducted further experiments in the revised manuscript (See Table 2 in Section 5.4). The results show consistent trends across these approaches, highlighting the generalization capacity of SynHING in generating HINs.
>
> [S2] We appreciate the reviewer for raising questions related to node features. Indeed, various node types in HIN can have their own features. It is possible to also generate attribute information for the non-target nodes., However, non-target nodes normally do not have their rich attribute information in existing real-world open datasets. Instead, they are commonly preprocessed by either constants, node IDs, or propagated features (Lv et al., 2021). For example, the commonly used IMDB dataset includes four node types: Movie, the target node type, which possesses rich feature information, and three minor node types: Director, Actor, and Keyword. The Director and Actor nodes typically use the one-hot encoding of human IDs as features. The Keyword node uses a bag-of-words approach, which is also represented using one-hot encoding for its features.  Such an approach has also been applied in other open HIN datasets including ACM and DBLP. Therefore, we follow this common setting, using node ID or node type information as node features for minor nodes to approximate these real-world datasets.
> We included statements to clarify this in the revised manuscript.

---

> > ### Author Response · Authors · 2024-11-28
> >
> > To further demonstrate the effectiveness of SynHING, we conducted additional experiments involving ablation studies (see Table 1) and implemented another meta-path-based HGNN, called HAN. The results of these experiments align with trends seen in other models, reinforcing the generalization capability of SynHING in generating HINs.
> >
> > Additionally, we conducted an extra pretraining/finetuning experiment with SimpleHGN. The updated Table 2 shows that the experimental trends of SimpleHGN are consistent with those of HGT and exhibit positive transfer effects across different datasets and HGNNs, confirming our experimental results.
> >
> > To showcase the contribution of SynHING to advancing interpretable machine learning, we embarked on further experiments employing the HGNN explainer. For our initial tests, we input synthetic ACM and DBLP into the xPath framework [1], which is a GNN explainer specifically designed for HINs. In the training phase, we use HGT as the backbone prediction model, which can effectively converge and achieve solid performance. During the explanation stage, xPath can successfully generate an explanation subgraph with decent accuracy and probability fidelity, as presented in the updated Section A.9 Implemented on HGNN Explainer.
> >
> > The additional results further demonstrate SynHING’s generality and practicality, addressing the scarcity of heterogeneous graph datasets and overcoming the need for such datasets in the domain of GNN explanations.
> >
> > Ref:
> >
> > [1] Tong Li, Jiale Deng, Yanyan Shen, Luyu Qiu, Huang Yongxiang, and Caleb Chen Cao. Towards fine-grained explainability for heterogeneous graph neural network. Proceedings of the AAAI Conference on Artificial Intelligence, 37(7):8640–8647, Jun. 2023b. doi: 10.1609/aaai.v37i7.

---

> > > ### Author Response · Authors · 2024-12-04
> > >
> > > We have carefully reviewed and addressed the comments provided by the reviewers. We hope our revisions have successfully addressed your concerns and met your expectations.

---

### Meta-Review · Area_Chair_23ZU · 2024-12-20

**Metareview:**

The paper introduces a framework for generating synthetic heterogeneous information networks (HINs) to enhance graph neural network (GNN) learning and model interpretability. The framework constructs HINs by identifying core structural motifs in real-world graphs, and merging them using controlled intra-cluster and inter-cluster methods to ensure realistic structures and statistical alignment with reference datasets. While the reviewers acknowledged the strong motivation behind the work, concerns persist regarding the empirical validation of its claims and methodology. Specifically, the reviewers felt the paper falls short in demonstrating scalability and lack of sufficient explainability experiments to validate the motivation.

**Additional Comments On Reviewer Discussion:**

The paper has received divergent reviews. While two reviewers have a favorable view of the work, two reviewers do not believe the quality to be of an acceptable level. In the post-rebuttal discussion phase, the reviewers were prompted to further discuss and share their final thoughts on acceptance and rejection. It emerged from this discussion that several concerns persist despite the rebuttal.

---

### Decision · Program_Chairs · 2025-01-22

Reject